# An In-Depth Review of the Benefits of Antibiotic Use in the Treatment of Borreliosis in Pregnancy

**Michael J. Cook** [1], **David Moynan** [2], **Gordana Avramovic** [2] and **John S. Lambert** [2,3,4,*]

1   Vis-à-Vis Symposiums, Capers End, Bury St Edmonds, Suffolk IP30 0AY, UK; mcook98@msn.com
2   Mater Misericordiae University Hospital, 44 Eccles St., D07 AX57 Dublin, Ireland
3   Catherine McAuley Centre, University College Dublin, 21 Nelson St., D07 A8NN Dublin, Ireland
4   Rotunda Maternity Hospital, D01 P5W9 Dublin, Ireland
*   Correspondence: jlambert@mater.ie; Tel.: +353-1-716-4530

**Abstract:** Evidence of congenital transmission of Lyme disease from mother to fetus has been investigated since the 1980s. This study reports the results of a retrospective analysis of 31 studies described in 27 papers published in a 2018 review article. Analysis of these identified statistically probable transmission in 13 (42%) of the 31 studies and possible transmission in 2 (6%). Most studies included mothers who had been treated with antimicrobials. When no antimicrobials were used, 74% of the pregnancies had adverse outcomes. When oral antimicrobials were used, 29% of pregnancies had adverse outcomes. When intravenous (IV) antimicrobials were used, adverse outcomes dropped to 12% of the pregnancies. This is a six-fold reduction in risk compared with no antimicrobial treatment. Some studies did not define whether oral or IV antimicrobials were used. When these results were included, adverse outcomes were 30 times higher for untreated mothers. Adverse outcomes included deaths, heart anomalies, and preterm births.

**Keywords:** congenital transmission; Lyme disease; Lyme borreliosis; birth outcomes; pregnancy; antimicrobials; transmission





## 1. Introduction

Lyme disease (LD) is a common zoonotic disease in many parts of the world. It is a vector-borne infection that is transmitted to humans by the bite of an infected tick. The potential for maternal–fetal transmission of *Borrelia* spp. has been discussed since the 1980s. Prior to the identification of LD, Lampert et al. in 1975 reported cases of infantile illness with clinical syndromes consistent with those of LD. Among these cases was an infant whose clinical course progressed from an erythema marginatum rash in the neonatal period to a painful and swollen right knee, among other symptoms, over the course of months [1]. A paper by Schlesinger et al., published in 1985, was the first to link fetal death with LD [2].

Analysis of a group of papers related to the issue of transmission of Borrelia bacteria from mother to fetus was used as the database for this study. The results are presented, with the evidence listed as possible or probable congenital transmission. Information regarding the impact of antimicrobial use in pregnant women on pregnancy outcomes is also presented.

## 2. Materials and Methods

An existing list of manuscripts extracted from a comprehensive literature review related to congenital transmission (CT) of Borrelia infections was used as the source of data on the congenital transmission of Lyme disease [3]. There were 27 papers in English related to CT. Data analysis on congenital transmission and treatment was further conducted. The number of adverse pregnancy outcomes and total cases were extracted from each study, and the probability of adverse events was compared with the study control group. Where the study did not have a control arm, the level of adverse pregnancy outcomes was compared to

general population studies in each specific country and time frame. The evidence presented in each study was assigned to one of 3 categories: (A) probable congenital transmission, (B) possible transmission, (C) no or low evidence of congenital transmission. A study was assigned to category (A), probable CT, if there was clear and positive proof of transmission or a probability of 90% or greater for CT was demonstrated. A study was assigned to category (B), possible transmission, if the probability of CT was lower than 90% but still statistically significant. A study was assigned to category (C), no or low evidence, if the authors indicated that an infant was "healthy" or the evidence was not statistically significant. The statistical method used for analysis was based on the Wilson score binomial interval calculation of the confidence interval [4]. Confidence intervals were generated using various confidence levels. When the confidence level equaled or exceeded 90%, the study result was defined as "probable transmission." When the confidence level was greater than 50% and less than 90%, the study result was classed as "possible transmission" [5].

## 3. Results

### 3.1. Evidence of Congenital Transmission

The data from each of the 27 papers are shown in Table 1. In some studies, the expected level of adverse events is known, for example, those studies in which *Borrelia* bacteria were found in the newborn. There would be no possibility of an infected tick bite for the fetus, and hence, transmission must have been from mother to fetus. Where the study had a control arm, the result is listed, and where there was no control arm, the incidence of adverse pregnancy outcomes was obtained from published studies, where available, for that region/location and time frame and listed as the "expected" level of adverse events. Explanatory notes and references to data sources are shown below the table.

**Table 1.** Evidence of congenital transmission; results of analysis of the 27 papers.

| Study | | Evidence | Total Cases | Adverse Outcome | Adverse % | Treated | Study CI | Control Mean or Expected | Note |
|---|---|---|---|---|---|---|---|---|---|
| Schlesinger 1985 | [2] | Probable | 1 | 1 | 100% | No | Single case | 0.0% | 1 |
| MacDonald 1986 | [6] | Probable | 4 | 4 | 100% | No | CI 95% (51–100%) | 0.0% | 1 |
| Markowitz 1986 | [7] | Possible | 6 | 2 | 33% | No | CI 40% (24.2–44%) | | |
| | | Low/No | 13 | 3 | 23% | Yes | CI 95% (8.2–50.3%) | | |
| Webber 1986 | [8] | Insufficient data | | | | | | | |
| MacDonald 1987 | [9] | Probable | 1 | 1 | 100% | No | Single case | 0.0% | 1 |
| Lavoie 1987 | [10] | Probable | 1 | 1 | 100% | No | Single case | 0.0% | 1 |
| Mikkelse 1987 | [11] | No/Low | 1 | 0 | 0% | Oral | Single case | 0.0% | |
| Weber 1988 | [12] | Probable | 1 | 1 | 100% | Oral | Single case | 0.0% | 1 |
| Williams 1988 | [13] | Probable | 255 | 26 | 10.2% | ? | CI 95% (7.1–14.5%) | 2.4% | 2 |
| Carlomango 1988 | [14] | Probable | 49 | 6 | 12.2% | 50% No | CI 90% (6.5–22%) | 6.1% | 3 |
| MacDonald 1989 | [15] | Probable | 8 | 8 | 100.0% | No | CI 95% (67.6–100%) | 0.0% | |
| | | No/Low | 1 | 0 | 0% | Oral | Single case | | |
| | | No/Low | 1 | 0 | 0% | IV | | | |

**Table 1.** *Cont.*

| Study | | Evidence | Total Cases | Adverse Outcome | Adverse % | Treated | Study CI | Control Mean or Expected | Note |
|---|---|---|---|---|---|---|---|---|---|
| Nadal 1989 | [16] | Probable | 12 | 7 | 58% | No | CI 95% (25.4–74.6%) | Less than 58% | 4 |
| Schutzer 1991 | [17] | No/Low | 1 | 0 | 0% | ? | Single case | 0.0% | 2 |
| Bracero 1992 | [18] | No/Low | 6 | 0 | 0% | Oral/IV | | | |
| Strobino 1993 | [19] | Possible | 10 | 3 | 30% | No data | CI 75% (16.5–48.2%) | 16.6% | 5 |
| Williams 1995 | [20] | Probable | 2386 | 31 | 1.3% | Yes | CI 95% (0.9–1.8%) | 0.5% | |
| Trevisan 1997 | [21] | Probable | 1 | 1 | 100% | No | Single case | 0.0% | |
| Maraspin 1999 | [22] | Low/No | 105 | 12 | 11% | IV | CI 95% (6.7–18.9%) | 10.3% | 6 |
| Strobino 1999 | [23] | No/Low | 796 | 4 | 0.50% | Yes | CI 10% (0.5–0.5%) | 0.6% | |
| Schauman 1999 | [24] | No/Low | 2 | 0 | 0% | IV | CI 10% (0–0.8%) | 0.0% | |
| Grandsaerd 2000 | [25] | No/Low | 1 | 0 | 0% | IV | CI 95% (0–79.3%) | 0.0% | |
| Gardener 1985 | [26] | | | | | | | | 7 |
| Walsh 2007 | [27] | No/Low | 1 | 0 | 0% | Oral | Single case | 0.0% | |
| Lakos 2010 | [28] | Probable | 10 | 6 | 60% | No | CI 95% (31.3–83.2%) | | |
| | | Probable | 19 | 6 | 31.6% | Oral | CI 95% (15.4–54%) | | |
| | | No/Low | 66 | 8 | 12.1% | IV | CI 95% (6.3–22.1%) | | |
| Maraspin 2011 | [29] | No/Low | 7 | 0 | 0% | IV | CI 95% (0–35.4%) | 0.0% | |
| Moniuszko 2012 | [30] | No/Low | 1 | 0 | 0% | Oral | | | |
| O'Brian 2017 | [31] | No/Low | 1 | 0 | 0% | IV | Single case | 0.0% | |

Notes and references to source data. Where treatment data is unknown a question mark is used (?), (No) is used where there was no treatment given, (Yes) denotes that treatment was given and this is detailed in Table 4 and (IV) denotes use of Intravenous antimicrobials detailed in Table 4. (Oral) denotes the use of oral antimicrobials. Note 1, transmission by tick bite to fetus is not possible. Note 2, No data listed for treatment of mothers. Note 3, *B. burgdorferi* IgG antibodies detected in the cord blood. Note 4, Authors state: seropositive women were more frequently (12.2%) detected among the spontaneous abortion group than among term pregnancy group (6.12%). Note 5, the abstract states six children affected; the text states seven children affected (58%). Note 6, Table II shows 11 women with positive titers, but text states data on congenital defects were available for 10 of the women with positive titers. The control group's adverse outcomes were 250 of 1510 (16.6%). Note 7, this is a book chapter [26].

The study results are summarized in Table 2.

### 3.1.1. Probable Transmission Studies

In 13 studies (42%), there was a statistical probability of 90% or greater that congenital transmission had occurred. The earliest study in 1985 by Schlesinger et al. [2] reported Lyme spirochetes in the spleen, kidneys, and bone marrow and stated that a teratogenic (structural) effect of the spirochetes could be possible. The three studies by MacDonald

between 1986 and 1989 all demonstrated Borrelia spirochetes in fetal tissues using immunofluorescence at autopsy [6,9,15]. Weber [12] used monoclonal antibody staining to detect *Borrelia burgdorferi*, and Lavoie [10] used culture and staining to identify *Borrelia*-like spirochetes. Since transmission occurred in the womb, there is no possibility of an infected tick bite, and the transmission must have occurred from the mother.

**Table 2.** Evidence of congenital transmission from the 31 studies described in 27 papers.

| Study Analysis | Number of Studies | % |
|---|---|---|
| Probable transmission | 13 | 42% |
| Possible transmission | 2 | 6% |
| No/low evidence | 16 | 52% |
| Total | 31 | 100% |
| Probable/possible | 15 | 48% |

In the relatively large study by Williams [13], there were 30 newborns with positive serology for *Borrelia burgdorferi.* Of these, 26 (10.2%) occurred in mothers from a Lyme endemic area and 4 (2.4%) in a nonendemic area. The infection rate for the nonendemic area was outside the 95% CI for the endemic area (95% CI 6.8–14.6%). This was highly significant. Carlomagno et al. [14], investigated the seroprevalence of Borrelia in a group of 49 patients with whom spontaneous abortions occurred and compared the results to the seroprevalence in a group of 49 mothers whose pregnancies went to term. The seropositive rate in the adverse outcome group was 12.2%, and in the successful pregnancies was 6.1%. The confidence level for a significant difference was 90%, and the study was assigned to the pregnancies "probable transmission" group.

Nadal et al. [16], in a serological survey over a 1-year period, included an investigation of cord blood using 12 mothers that were seropositive for *B. burgdorferi*. Adverse outcomes occurred in seven cases (58%). An adverse event level of much less than 58% would be expected.

A paper by Williams et al., published in 1988, identified 26 of 255 (10.2%) infants from an endemic area seropositive for *B. burgdorferi* and 4 of 166 (2.4%) infants seropositive from a nonendemic area. The increased risk of transmission in the endemic area was statistically significant at the 95% level. The authors did not find a statistical difference in adverse outcomes, which was to be expected with the small sample size of one malformation in the endemic group and 0 (zero) in the nonendemic group. A second study by Williams et al., published in 1995 [20], compared adverse outcomes of pregnancies in a group of 2386 newborns in an LD endemic area and 2428 newborns in a nonendemic area. There were no statistical differences for the majority of reported outcomes, but congenital heart disease was 12.99 per 1000, or 1.3% (CI 95% 0.9–1.8%), in the endemic area and 5.35 per 1000, or 0.54% (CI 95% 0.31–0.91%), in the nonendemic area. This was significant at the 95% confidence level, with the nonendemic mean outside the range for the endemic area.

Trevisan et al. [21] reported on a case in which, 3 weeks after birth, an erythema appeared, and regression and relapse occurred with more serious clinical symptoms appearing. At 9 months, investigations identified spirochetes in the epidermis and dermis, and blood samples were seropositive for IgG antibodies. The infant received five courses of antibiotics over a 3-year period. Although it was possible the baby suffered a tick bite within the first few weeks after birth, this probably represented a case of congenital transmission.

### 3.1.2. Possible Transmission Studies

Markowitz et al. [7] reported adverse events in 2 of 6 cases (33%) in which the mother did not receive treatment and in 3 of 13 cases (22%) in which the mothers were treated with antibiotics. This was statistically significant at the 40% confidence level and classed as possible transmission. Strobino et al. (1993) [19] identified 3 of 10 (30%) congenital defects in mothers who were seropositive and 16.6% congenital defects in 250 out of 1510 women who were seronegative and with a negative clinical history of LB. There was a 75% confidence

level for transmission with an adverse outcome for infected mothers compared with the control group.

### 3.2. Antimicrobial Use and Adverse Events Data

Most studies included mothers who were treated with antimicrobials during their pregnancy. The study by Lakos et al. [28], for example, included mothers who had various treatment protocols, including intravenous (IV) penicillin or ceftriaxone in which there were 8 of 66 (12.1%) AE, oral antimicrobials including multiple courses, and 10 women who were not treated. The Markowitz et al. study [7] included 12 mothers who were treated with IV ceftriaxone. A summary of the antimicrobial regimens and outcomes are shown in Table 3. These were extracted from all papers in which data were available and listed in Table 4. Many studies clearly defined the treatment protocol used, and these were assigned to three groups: intravenous antimicrobials (IV), oral antimicrobials, and no treatment.

**Table 3.** Adverse outcomes where full treatment data were available.

| Antimicrobial Treatment | Study Cases | Adverse Outcomes | Adverse Outcome % | Risk Factor | Confidence Interval (CI) |
|---|---|---|---|---|---|
| IV | 196 | 23 | 12% | 1 | CI 95% (7.6–17.1%) |
| Oral | 24 | 7 | 29% | 3 | CI 95% (12.6–51.1%) |
| Untreated | 31 | 23 | 74% | 6 | CI 95% (55.5–88.1%) |

**Table 4.** Studies that included data on maternal antimicrobial treatment.

| Study | Maternal Treatment | Number of Cases | Adverse Outcomes |
|---|---|---|---|
| Schlesinger 1985 | Untreated | 1 | 1 |
| MacDonald 1986 | Untreated | 4 | 4 |
|  | IV penicillin | 12 | 2 |
| Markowitz 1986 | Treatment not defined | 13 | 3 |
|  | Untreated | 6 | 2 |
| MacDonald 1987 | Untreated | 1 | 1 |
| Lavoie 1987 | Untreated | 1 | 1 |
| Mekkelsen 1987 | Oral penicillin | 1 | 0 |
| Weber 1988 | Oral penicillin | 1 | 1 |
| Williams 1988 | Endemic area seropositive | 255 | 26 |
|  | Nonendemic area | 111 | 4 |
| Carlomango 1988 | 1 yes, 3 no, 2 unknowns | 12 | 6 |
|  | IV penicillin as per syphilis | 2 | 0 |
| MacDonald 1989 | Oral | 1 | 0 |
|  | Untreated | 8 | 8 |
| Nadal 1989 | No | 12 | 6 |
| Schutzer 1991 | No information | 1 | 0 |
| Bracero 1992 | Oral/IV physician choice | 6 | 0 |
| Strobino 1993 | No information | 10 | 3 |
| Williams 1995 | Treatment not defined | 2386 | 31 |
| .Strobino 1999 | Treatment not defined | 796 | 4 |
| Bracero 1992 | Oral/IV physician choice | 6 | 0 |
| Trevisan 1997 | Not during pregnancy | 1 | 1 |
| Grandsaerd 1999 | IV ceftriaxone 2 g/day 14 days | 1 | 0 |

**Table 4.** *Cont.*

| Study | Maternal Treatment | Number of Cases | Adverse Outcomes |
|---|---|---|---|
| Maraspin 1999 | IV ceftriaxone 2 g/day 14 days. Note 2 | 105 | 12 |
| Schauman 1999 | IV ceftriaxone 2 g/day 14 days | 2 | 0 |
| Walsh 2007 | Oral amoxycillin 500 mg t.i.d. | 1 | 0 |
|  | IV ceftriaxone 2 g/day 15 days | 66 | 8 |
| Lakos 2010 | Oral 20 days or more Ceftriaxone? | 19 | 6 |
|  | Untreated | 10 | 6 |
| Maraspin 2011 | IV ceftriaxone 2 g/day 14 days | 7 | 1 |
| Moniuszko 2012 | Oral amoxycillin dose not specified | 1 | 0 |
| O'Brian 2017 | IV ceftriaxone 2 g/day 1 day | 1 | 0 |

The risk of adverse events was significantly lower when the mother was treated during pregnancy. Treatment using IV antimicrobials resulted in the lowest level of adverse outcomes. When oral antimicrobials were used, the adverse outcomes were three times greater and the risk six times greater when the mother was not treated.

In a further six studies, mothers were treated with antimicrobials, but the protocols were not specified. The Williams et al. 1995 study included 2386 cases [20], and only 31 (1.3%) pregnancies had adverse outcomes. Strobino et al. [23], in their 1999 paper, had 796 cases in which mothers were treated with antibiotics and only 4 (0.5%) adverse outcomes. When these data were included with the data from known antimicrobial protocols, there were only 2.4% of cases with adverse outcomes compared with 74.2% for untreated mothers. This indicates that when mothers are untreated, the risk of adverse events is 30 times greater than when mothers are treated with antimicrobials (see Table 4). In six of the studies, there was no specific information regarding the class of antibiotic or duration. Despite this lack of key data in some of the papers, there is significant evidence of the benefits of antibiotic treatment, as shown in Table 4. However, the benefits of oral versus IV antibiotics should be evaluated in greater depth.

A summary of all data for antimicrobial treatment and outcomes is shown in Table 5.

**Table 5.** Summary of antimicrobial treatments.

| Antimicrobial Treatment | Study Cases | Adverse Outcomes | Adverse Outcome % | Risk Factor | Confidence Interval (CI) |
|---|---|---|---|---|---|
| Method known | 220 | 30 | 13.6% |  | CI 95% (9.4–18.9%) |
| Method unknown | 2405 | 34 | 1.4% |  | CI 95% (1–2%) |
| Total treated | 2625 | 64 | 2.4% | 1 | CI 95% (1.9–3.1%) |
| Untreated | 31 | 23 | 74.2% | 30 | CI 95% (55.4–88.1%) |

## 4. Other Tickborne Diseases

Whilst the impact of Borrelia infection on pregnancy has significant coverage, as discussed here, there are many other tickborne infections that can cause disease in humans. Lambert discussed ehrlichiosis, babesiosis, relapsing fever, and other diseases that can result from tick bites and recommended actions to improve the understanding of tickborne infection in pregnant women and their infants [32]. These can be encountered as the predominant disease or present in a patient in conjunction with Lyme disease. Thomas et al.

investigated the influence of dual infection with the agents for Lyme disease and ehrlichiosis [33]. Larsson et al. described transplacental transmission and pregnancy complications caused by tickborne relapsing fever (TBRF), including miscarriage and neonatal death in sub-Saharan Africa [34]. Jakab et al., in a recent review, provided extensive data on the various TBRF Borrelia species and their prevalence around the world [35]. The majority of TBRF disease is carried by soft-bodied ticks, with *B. miyamotoi* carried by hard-bodied *Ixodes* species ticks and is present in North America and Europe. The US Centers for Disease Control and Prevention reported the clinical course of a mother and child with TBRF and stated that it could pose serious risks to the mother and neonates [36]. An early report of simultaneous infection of a human with Lyme disease and babesiosis was published by Benach et al. in 1985 [37]. They found that 54% of patients with babesiosis tested positive for Lyme disease. A comprehensive study of pathogens that coinfect humans was published by Bergfoff [38] and indicated that there was evidence that coinfections could exacerbate Lyme disease. The impact on the fetus and infant when the mother has multiple coinfections with tickborne or other diseases should be investigated.

## 5. Discussion

None of the studies analyzed in this review were randomized controlled trials (RCTs), the gold standard for examining the efficacy of medical interventions. However, there are ethical considerations that prevent the investigation of congenital transmission of Lyme disease using this method. This leaves alternatives such as observational and retrospective cohort studies, as reported here. They are frequently considered less reliable, but comparisons of RCTs with observational studies have indicated that the two methods can give consistent results, and observational studies are recommended for use when there are issues related to RCT [39,40].

The authors of most papers gave specific recommendations for the treatment of mothers. Nine papers recommended prompt treatment. A large majority of the papers were published more than 20 years ago, and only two papers that each described one case were published in the last 10 years. Antimicrobial therapy, to be of benefit, requires an accurate diagnosis of Lyme disease. There is evidence that the actual incidence of Lyme disease is far greater than official data suggest. For example, in the United States, approximately 30,000 cases are reported to the Centers for Disease Control and Prevention (CDC) [41], but a study carried out using data from seven testing laboratories gave an estimate of 240,000 to 444,000 infected patients in 2008 [42], and a recent report by the CDC using health insurance claims information generated an estimate for annual cases of Lyme disease of 405,000 to 547,000 over the period from 2010 to 2018 [43]. These estimates are based on diagnosed and treated cases and do not include those people who were undiagnosed or misdiagnosed.

In England and Wales, approximately 1000 cases per year are reported by Public Health England. However, a recent study estimated that there were potentially more than 100,000 cases in 2018 in England and Wales and 10,000 cases in Ireland, with a prevalence of 49,000 (not accounting for Northern Ireland) [44]. Moreover, the testing methods, including Western blot, enzyme-linked immunosorbent assays, and a combination of two tests (two-tier methodology) recommended by most health authorities, have poor sensitivity, as summarized in two papers by Cook/Puri [45,46]. This suggests considerable underdiagnosis of LD and, hence, a lack of treatment, with the consequence being a significant ongoing risk to babies and pregnant mothers.

In addition to the statistical analysis of the studies, they also contained specific evidence of maternal–fetal transmission. Lakos et al. [28], found that all IgG-antibody-positive mothers gave birth to IgG-antibody-positive infants. Typically, days to weeks after infection, humans generate IgM antibodies, and IgG antibodies develop weeks to months thereafter. The presence of IgG antibodies could indicate that the infants had been infected weeks to months prior to birth. However, whereas IgM and IgG antibody production can be used to indicate active infection in a newborn, the kinetics of IgM are poorly understood

with LD. A study by Aberer et al. indicated that an IgM and IgG response was detected within 2 weeks of a tick bite in some patients, and detection of both IgM and IgG varied significantly over a 10-week period [47]. In addition, this paper reported that 12 (20%) of the patients were constantly IgM seropositive after treatment.

Objective evidence of fetal *Borrelia* infection was found in multiple papers [6,15,20], with *Borrelia* spp. identified in tissues from viscera, including the liver, heart, and brain, through immunofluorescence staining, dark field microscopy, or Warthin–Starry staining. As discussed previously, Lyme disease is a zoonosis that is acquired through the bite of a *Borrelia*-infected tick. Since there is no possibility of a fetal tick bite in utero, we believe that the identification of *Borrelia* in fetal tissues must be considered categoric proof of congenital transmission. Markowitz et al. [7] identified two cases of premature/fetal death in their study (10.5%). Statistical analysis suggests that there is a 75% probability of CT; conservatively, the study was classed as "possible transmission." Where infection is identified within 2 weeks of a tick bite, the current standard of care for the pregnant mother and unborn child in the UK is defined by the National Institute for Health and Care Excellence. The guideline recommendation is to treat Lyme disease in pregnancy the same as for nonpregnant people, with the majority of conditions using oral antibiotics, or for cases with involvement of the central nervous system, intravenous ceftriaxone. If the newborn is IgM-positive or infection is suspected, treatment under specialist care is recommended [48]. The US Centers for Disease Control and Prevention recommend that pregnant mothers should be treated as nonpregnant people using oral antibiotics [49]. The European Centre for Disease Prevention and Control website states that most cases can be successfully treated with a few weeks of antibiotic treatment. The authors recommended immediate treatment with penicillin for any mother who acquired LD during pregnancy, with IV penicillin for late-stage illness. Two recently published papers were not in the main group. In a cross-sectional survey by Leavey et al. that included 691 eligible mothers, there was evidence that antimicrobials reduced neonatal pathology, including rashes, hypotonia, and respiratory disease [50]. The authors considered the results provided evidence of the need for rigorous prospective observational studies to minimize bias associated with participants living in Canada or the United States. Trevisan et al. monitored pregnancies at an Italian hospital in Trieste from 2008 until 2020. Eight women were seropositive for LB and were immediately treated with antibiotics (amoxicillin 1 g 3x/day for 14 days). There were no adverse events for either the mothers or the newborns [51]. The authors considered the sample size limited the ability to conclusively define the impact of treatment, and again recommended larger studies.

## 6. Conclusions

This study examined manuscripts related to the congenital transmission of Lyme disease, identified by Waddel et al [3], published between 1985 and 2017, and conducted further analyses on treatment. The majority (80%) were published more than 20 years ago. Of the 31 studies described in 27 papers with data on congenital transmission, 13 (42%) had statistically significant evidence (90% confidence level or higher) of maternal–fetal transmission of *Borrelia* infection, and a further 2 (6%) gave evidence of transmission, though with a lower confidence level. Detailed information on the antimicrobial treatment of the mother during pregnancy was defined in 14 studies. When IV treatment was used, 12% of the pregnancies had adverse events; with oral treatment, the risk doubled, with adverse outcomes noted in 29% of cases; and when mothers were untreated, the risk was six times higher, with 74% of pregnancies with adverse events. Some studies indicated that antimicrobials were used, but the specific protocols were not defined. When the data for these studies were included, the risk of adverse outcomes for untreated mothers was 30 times greater than treatment using any form of antimicrobials. The need for antibiotic treatment was specified in 11 papers, with prompt use specified by five groups of researchers. In the context of high numbers of undiagnosed and misdiagnosed LD

referenced earlier, data show that the congenital transmission of Lyme borreliosis has a significant impact on pregnancy outcomes and entails high risks of fetal injury and death.

**Author Contributions:** Conceptualization, M.J.C. and G.A.; methodology, M.J.C. and G.A.; statistical analysis, M.J.C.; writing—original draft preparation, M.J.C.; writing—review and editing, M.J.C., G.A., J.S.L. and D.M.; M.J.C. had full access to all of the data in the study and takes responsibility for the integrity of the data and the accuracy of the data analysis. All authors have read and agreed to the published version of the manuscript.

**Funding:** This research received no external funding.

**Data Availability Statement:** All data used in this study are available in the referenced documents.

**Acknowledgments:** We thank the reviewers for their work and suggestions that were used to improve the manuscript.

**Conflicts of Interest:** The authors declare no conflict of interest.

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
