# Peer review of "An In-Depth Review of the Benefits of Antibiotic Use in the Treatment of Borreliosis in Pregnancy"

_2673-8007, doi:10.3390/applmicrobiol3020022_

Round 1

Reviewer 1 Report

See attached Word file

Author Response

Many thanks to the Reviewer for the extensive work and suggestion. All suggestions have been implemented and the careless spelling and grammatical errors corrected

General Comments

                reference to Lambert, Fron Med. 2020;7:72 has been added in the text.

Major Comments

  • The reviewer has highlighted an error and also that that data needs clarification for calculations. There were 27 papers published, of which 13 (48%) demonstrated congenital transmission. Some papers described multiple samples/studies. For Example MacDonald 1989 reported three separate studies with 8 mothers untreated, 1 treated with oral antimicrobials and 1 treated using IV. Similar multiple studies were reported by Markwitz 1986, and Lacos 2010. When percentages are calculated using the number of studies that data is reported in Table 5.
  • Lines 94 to 108 should state the numbers of cases in the Williams paper. This has been done and additional wording added for emphasis.
  • Lines 130 and Table 4. This section has been modified as suggested by the Reviewer with emphasis on the benefits of treatment and the limitations of due to lack of antibiotic protocol details. As suggested the need for additional evaluation of oral versus IV is added.
  • Regarding the Lambert 2020 paper. The suggestion is to comment on the confounding variable (other tick borne diseases). This has been added along with a discussion regarding impact of other tickborne infections in pregnancies with the suggested references.
  • The newer articles have been included.

Minor Comments:

  • Numerous grammatical errors. Apologies for these. I believe all have been corrected.
  • Recommend “but”: Accepted now line 190.
  • Line 183: suggested shortening: Accepted now lines 201/2
  • Line 195: “...detected withing 2 week of a tick bite”. Need a reference for persistent IgM positivity in chronic Lyme disease. The following statement has been added: Also, this paper reported that 12(20%) of the patients were constantly IgM seropositive after treatment.
  • Line 212 Suggest: "If the new-born is IgM positive or infection is suspected, treatment under specialist care is recommended although a specific protocol is not defined" This has been reworded to indicate more clearly that the guidelines do not recommend a specific protocol. This is now Line 230.

Reviewer 2 Report

Novel approach of review of the still limited number of reports of Lyme & pregnancy outcomes, a type of meta-analysis.  Also, limited by some incomplete data in the primary sources.  Nonetheless, some useful data is 'extractable' and the authors make the case for a substantial impact of Lyme disease in pregnancy at least partially mitigatable by application of antimicrobial treatment, albeit of mostly rather limited durations and the contrast between outcomes with NO antimicrobial treatment, oral antimicrobial treatment and intravenous antimicrobial treatment.  The work compels formal, systematic prospective and or observational longitudinal studies of women who become pregnant with pre-existing Lyme disease or acquire Lyme disease during pregnancy.  Long term follow up of both mother and infants and scientific study using a wide range of methods, including 'cutting edge' technologies are long overdue.  Advocates for this type of study are beginning to be 'heard' in the 'corridors' of Power.  The study is timely.

One caveat:  the Reviewer has little formal training in statistical methods and suggests to the Editors that at least one reviewer well-versed in statistical methods review the data analyzed and conclusions drawn by the authors. 

The current reviewer is 'taking at face value' the authors' data analyses.

A few specific issues in the mss.:

Table 4: under Schauman 1999, drug is not specified.  Presumably the report is on ceftriaxone 2 g/day 14 days.

Line 180-181: "..with a prevalence of 49 000...."  Is that what is meant?  Or is it 49 per 1000 ?

Line 195-196: "...detected within 2 weeks of a tick bite...."

Lines 237-238:   "...Lyme borreliosis has a significant impact on pregnancy outcomes and entails high risk of fetal injury and death."

Reference 8 & 30 are duplicative and ARE the same citation/reference. So this needs to be corrected in the Refs and in the citations in the text of the mss. 

Author Response

Many thank to the Reviewer for the observations and suggestions. All have been implemented.

Table 4 drug not specified:  This is corrected with ceftriaxone added.

Line 180-181:  A comma has been added to indicate thousands, and also added on the prior page.

Line 195-196: The suggested words are added (Now line 224)

Line 237 238:  The word “entails” has been added to clarify (Now line 261)

Reference 8 & 30 are duplicative:  This has been corrected and the duplicate Ref 30 deleted

Round 2

Reviewer 1 Report

The authors have done a good job responding to the reviewer comments. Several minor problems remain.

1. Abstract, Table 2 and Conclusion: The numbers do not match (48% vs 42%). The authors should change the Abstract (line 11) and Conclusion (line 266) and say "31 studies described in 27 papers", as shown in Table 2. Then the numbers in Table 2 and the Conclusion should be used in the Abstract (probable transmission 42%, possible transmission 6%) and all three would match.

2. Several grammatical errors remain: "The increased risk..." (line 110); "which is to be expected" (line 112); "Although it is possible that the baby suffered a tick bite within the first few weeks after birth, this probably represents a case of congenital transmission" (line 124); "Most studies included mothers who..." (line 136); "The Markowitz et al study [14] included 12 mothers who..." (line 140); "many other tickborne infections" (line 168); "A comprehensive study of pathogens that co-infect humans with Borrelia..." (Line 184); "Nine papers recommended prompt treatment" (line 198); "If the newborn is IgM positive or infection is suspected, treatment under specialist care is recommended" (line 243); "In a cross sectional survey by Leavey et al that included..." (line 251); "There were no adverse events for either the mothers or the new-borns" (line 258). 

3. Use "but" instead of "however" (lines 78, 116, 150, 193, 245, 273)

Author Response

Response to Reviewer 1 Round 2.

Many thanks again for the thourough analysis and inputs. There were 48% of papers that showed transmission and 42% of studies within those papers. To avoid confusion the suggestion to use the 31 studies and 42% of those has been adopted, leaving the correct value 6% of 31 studies showing possible transmission. Abstract, table 2 and conclusion now match. All other listed suggestions have also been been adopted.